# TANGO1 inhibitors reduce collagen secretion and limit tissue scarring

Ishier Raote [1,2] ✉, Ann-Helen Rosendahl [3], Hanna-Maria Häkkinen[1], Carina Vibe[1,4], Ismail Küçükaylak[5], Mugdha Sawant[3], Lena Keufgens[6], Pia Frommelt[7], Kai Halwas[5], Katrina Broadbent[1], Marina Cunquero [8], Gustavo Castro[8], Marie Villemeur[2], Julian Nüchel [9], Anna Bornikoel [3], Binita Dam[10], Ravindra K. Zirmire[10], Ravi Kiran [10], Carlo Carolis [1], Jordi Andilla [8], Pablo Loza-Alvarez [8], Verena Ruprecht [1,11,12], Colin Jamora [10], Felix Campelo [8], Marcus Krüger [6], Matthias Hammerschmidt [5,13], Beate Eckes [3], Ines Neundorf [7] ✉, Thomas Krieg[3,6,13] & Vivek Malhotra [1,11,12] ✉

Uncontrolled secretion of ECM proteins, such as collagen, can lead to excessive scarring and fibrosis and compromise tissue function. Despite the widespread occurrence of fibrotic diseases and scarring, effective therapies are lacking. A promising approach would be to limit the amount of collagen released from hyperactive fibroblasts. We have designed membrane permeant peptide inhibitors that specifically target the primary interface between TANGO1 and cTAGE5, an interaction that is required for collagen export from endoplasmic reticulum exit sites (ERES). Application of the peptide inhibitors leads to reduced TANGO1 and cTAGE5 protein levels and a corresponding inhibition in the secretion of several ECM components, including collagens. Peptide inhibitor treatment in zebrafish results in altered tissue architecture and reduced granulation tissue formation during cutaneous wound healing. The inhibitors reduce secretion of several ECM proteins, including collagens, fibrillin and fibronectin in human dermal fibroblasts and in cells obtained from patients with a generalized fibrotic disease (scleroderma). Taken together, targeted interference of the TANGO1-cTAGE5 binding interface could enable therapeutic modulation of ERES function in ECM hypersecretion, during wound healing and fibrotic processes.

A major challenge in regenerative medicine is to optimize tissue regeneration by therapeutic modulation of its natural ability to form a scar after a mechanical injury or during disease processes. After tissue injury, platelets, endothelial cells, and inflammatory cells induce fibroblasts to initiate extracellular matrix (ECM) deposition. Early granulation tissue is abundant in fibronectin and other ECM constituents. Fibrillar collagens form the main component of the final ECM and scar tissue[1,2]. Pathological scarring can develop in most tissues and represents a major cause of death, impaired quality of life, and economic burden, especially in aging societies. No therapy is as yet available and there is an urgent need to uncover new targets to treat scarring and fibrosis[3].

Scleroderma (systemic sclerosis), an autoimmune-driven, complex disease leading to extensive fibrosis of many organs including

the skin is often used as a model disease to study principal mechanisms of fibrosis[4,5]. Initial causes are still unknown but the disease reflects general features of fibrotic processes leading to activation of fibroblasts and excessive deposition of a collagen-rich ECM[6].

In spite of significant progress in understanding the pathophysiology of the underlying processes of fibrosis and identifying the role of several fibrogenic cytokines, few specific therapeutic strategies are available to reduce or control connective tissue formation during wound healing and fibrosis[4]. The presence of several concurrent fibrogenic signaling pathways may explain why trials of specific inhibitors of interleukin 4 (IL-4) or TGFβ pathways alone have yielded disappointing results as antifibrotic targets[7,8]. However, all these fibrotic signal transduction cascades converge to produce at least one consistent hallmark of a broad range of pathologies—there is always excessive secretion of ECM proteins, such as collagen. Modulation of this excess ECM deposition could therefore offer a promising disease-modifying, therapeutic approach.

Targeted control of collagen deposition requires a detailed understanding of the mechanisms governing the secretion of these large proteins. Of particular importance is their export from the endoplasmic reticulum (ER), a process that has only recently become amenable to molecular analysis after the discovery of the TANGO1 family of proteins, including TANGO1 and its paralog cTAGE5[9–11]. TANGO1 is present in most metazoans and is required for collagen export from the ER in all animals tested thus far, including mammals (humans, canines, mice), Drosophila, and zebrafish[12–18].

TANGO1 family proteins have emerged as master organizers of ER exit site (ERES) machinery, building transient ER-ERGIC-Golgi inter-organelle tunnels to increase secretory capacity for bulky cargoes like collagen[19,20]. Given the essential role of TANGO1 family proteins in collagen secretion, inhibiting their function provides a likely approach to control pathological collagen hypersecretion. TANGO1 is a transmembrane protein that is resident at ERES where, through an ER-luminal SH3-like domain, TANGO1 recruits collagen. Via cytoplasmic interactions, TANGO1 organizes the ERES and the ERES-ERGIC interface[9,10,21–24]. Each of these interactions is required for TANGO1 function and collagen secretion; controlling any of them should provide a handle to regulate collagen secretion.[25]

To achieve this control, we have developed membrane-permeant peptide inhibitors of TANGO1, equivalent to regions of TANGO1 and cTAGE5 CC2 domains, conjugated to dodecapeptides that allow for their cytoplasmic delivery. Such cell-penetrating peptides (CPPs) have emerged as powerful tools for targeted intracellular protein delivery[26].

Biologists have traditionally considered ER export un-druggable because its machinery is essential for cell viability and acts constitutively, such that targeting any component could produce pleiotropic, off-target effects on secretion. However, recent studies identifying specific interactions among ERES machinery and secretory cargoes have hinted at the potential for targeted interventions[27,28]. Here, our precise targeting of the interface of ERES proteins, TANGO1 and cTAGE5, controls their activity specifically, showing that it is possible to control ER export of specific ECM proteins, without major off-target toxicity.

We show that the peptide inhibitors reduce collagen secretion in normal human fibroblasts, in activated fibroblasts, and in fibroblasts obtained from patients with scleroderma. Since major stages of cutaneous wound healing are conserved in zebrafish, we also used a zebrafish model of wound repair, where a laser is used to introduce wounds quickly and reproducibly[29] and demonstrated reduced scar formation upon treatment with peptide inhibitors.

Our results suggest a promising avenue for therapeutic intervention via targeted inhibition of ER export, for currently intractable fibrotic disorders characterized by excessive ECM production.

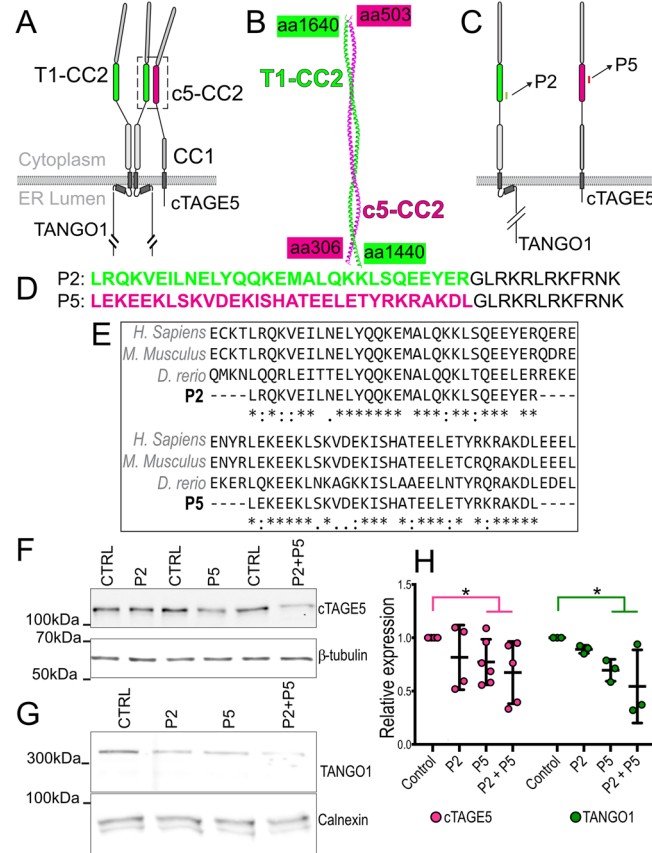

**Fig. 1 | Design of peptides to inhibit TANGO1-cTAGE5 heterodimerization. A** The organization and the domains of TANGO1 and cTAGE5. Each of these proteins contains two cytoplasmic coiled-coil domains. TANGO1 coiled-coil 2 (T1-CC2) is highlighted in green. cTAGE5 CC2 in magenta. **B** Alphafold2 prediction of the structure of the heterodimer of TANGO1 (green) and cTAGE5 (magenta). **C** Schematic indicating where the peptide sequences map to in TANGO1 and cTAGE5. **D** Two peptides were synthesized of 30 (P2) and 31 (P5) amino acids respectively, conjugated to a 12 amino-acid C-terminal lysosomal escape motif (shown in black text). **E** Alignment of human, mouse, and zebrafish sequences. **F** Representative western blot of U2OS cell lysates treated with P2, P5 or P2 + P5, probed for cTAGE5. β-tubulin was used as a loading control. **G** Representative Western blot of U2OS cell lysates treated with P2, P5 or P2 + P5, probed for TANGO1. Calnexin was used as a loading control. **H** Quantification of blots showing TANGO1 (green) and cTAGE5 (magenta) levels (mean +/- SD) in control or peptide-treated cells (normalized to control) *$p < 0.05$, Student's $t$ test comparing cTAGE5 (magenta line) or TANGO1 (green line) in control vs treated. $N = 3$ (TANGO1), $N = 5$ (cTAGE5).

## Results

### Peptides destabilized both TANGO1 and cTAGE5

Prior studies have revealed that TANGO1 heterodimerization with its paralog cTAGE5 is mediated by a coiled-coil region (shown as CC2 in the two proteins, Fig. 1A)[10]. We used AlphaFold2[30] to predict the structure of the TANGO1−cTAGE5 interface (T1-CC2−c5-CC2, Fig. 1B). The primary sequence used in the predictions is indicated by the amino acid numbering, corresponding to the numbers on Uniprot (Uniprot IDs, TANGO1: Q5JRA6, cTAGE5: Q96PC5-7). According to the predictions, the coiled-coil regions dimerize along their entire length (Fig. 1B, and Supplementary Fig. 1).

Based on these structures, we designed short polypeptides corresponding to the primary sequences of these coiled-coil regions of TANGO1 and cTAGE5, which should competitively inhibit the TANGO1-cTAGE5 function (Fig. 1C). We designed two peptides, named P2 and P5, of 30 or 31 amino acids respectively, conjugated to a

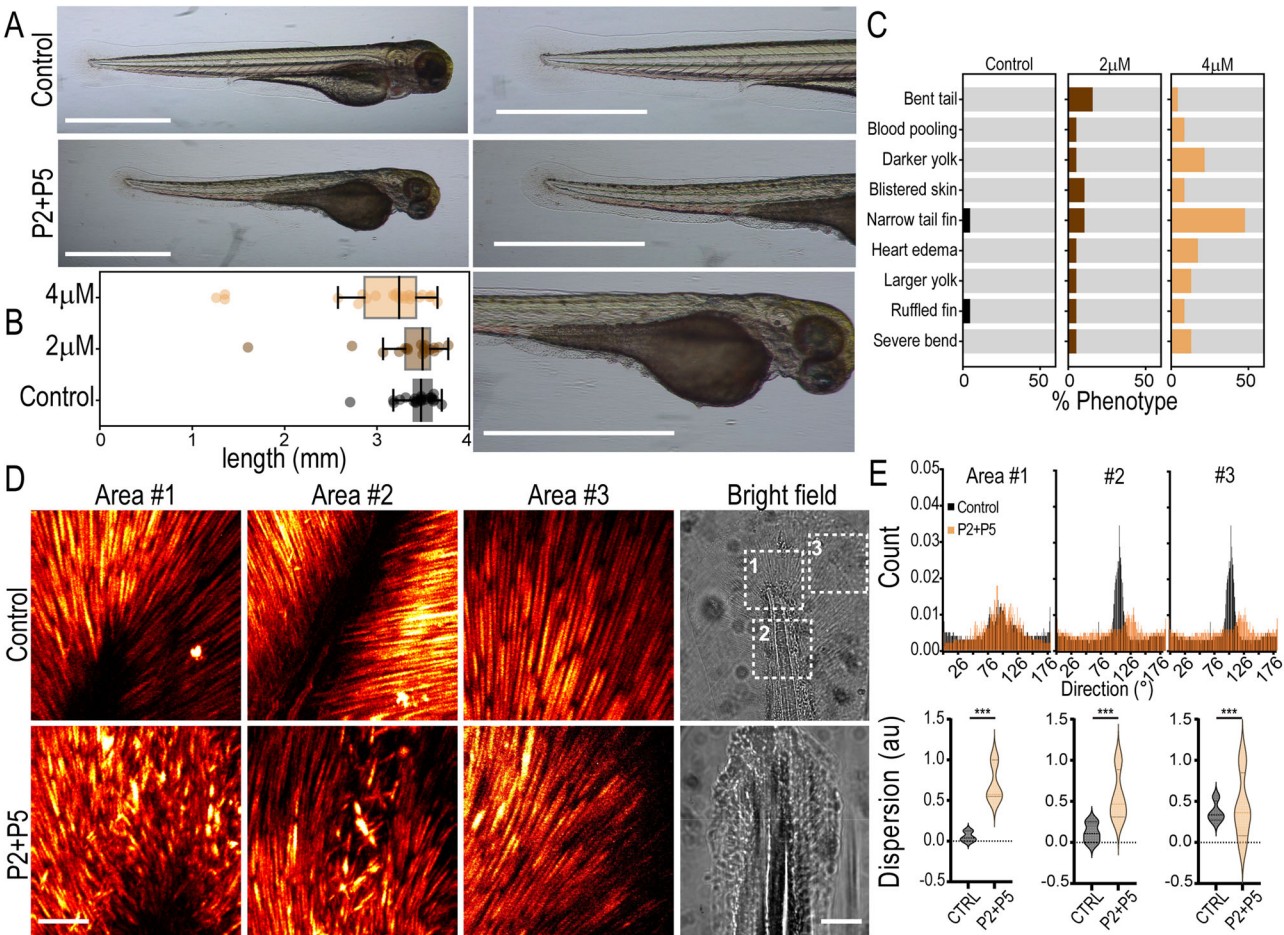

**Fig. 2 | Peptides induce phenotypic changes in zebrafish consistent with TANGO1/cTAGE5 inhibition. A** Zebrafish larvae at 3 dpf. Fish were treated with P2 + P5 (4 μM), or left untreated in E3 (control). Magnified images of the tail and yolk on the right. Scale bars 1 mm. **B** Quantification of larva length from head to tail (boxplot with median and interquartile range $N$ = 21, 19, 23 in control, 2 μM and 4 μM respectively). Whiskers connect the maximum and minimum.
**C** Quantification of phenotypes observed in fish for $n$ = 21, 19, 23 replicates in control, 2 μM and 4 μM respectively, from three independent experiments.
**D** Maximum projection of the entire Z-stack of SHG images from three different tail fin regions (Areas #1, 2, 3) as indicated in the bright-field images. Images were

acquired from three control and treated fish each, with approximately similar regions of the tail chosen as Area #1, 2, and 3. Scale bars 100 μm. **E** Quantification of collagen fiber direction obtained from the SHG signal in the three segmented areas approximately represented in (**D**) Top row: Collagen fiber direction histograms from a representative control or treated fish. Bottom row: Collagen fiber dispersion distribution plots from a representative control or treated fish. This measure represents the dispersion of the preferred orientation of collagen fibers in the image. Control measurements are represented in black and P2 + 5 (4 μM) is treated in orange. Kolmogorov–Smirnov test *** $p < 0.001$.

cell-penetrating dodecapeptide[31] (GLRKRLRKFRNK) to translocate the attached peptides into the cytoplasm (Fig. 1D) P2: LRQKVEILNE-LYQQKEMALQKKLSQEEYERGLRKRLRKFRNK and P5: LEKEEKLSKVDE-KISHATEELETYRKRAKDLGLRKRLRKFRNK. This stretch of amino acids in TANGO1 and cTAGE5 are well conserved, showing high identity even in zebrafish cTAGE5 (Fig. 1E). As controls, amino acids of P2 and P5 were scrambled to generate scrP2: VEEAKLYQRQSRMENLLEKEKQE LLQKQIYGLRKRLRKFRNK and scrP5: KLRKHELERLKSELEEIEATETKD YVADKKSGLRKRLRKFRNK.

We quantified levels of TANGO1 and cTAGE5 by western blotting in control cells and those treated with peptides P2 (100 μM), P5 (100 μM), and P2 + P5 (100 μM each). Both TANGO1 and cTAGE5 protein levels were reduced in the presence of P2 and P5 (Fig. 1F, G, quantified in H), suggesting that targeting the TANGO1/cTAGE5 heterodimerization interface leads to unstable monomers.

**Peptides inhibit TANGO1/cTAGE5 in zebrafish**
To test whether peptides inhibit TANGO1/cTAGE5 in vivo, we treated zebrafish embryos from the 2-cell stage with a combination of P2 and P5 (2 μM or 4 μM, each). Embryos were continuously treated with the

peptides until 3 days post fertilization (dpf), when they were imaged. At 3 dpf, larvae treated with peptides had a shorter body axis (Fig. 2A, B), a phenotype also seen in TANGO1/cTAGE5 zebrafish knockouts[16]. Some fish also presented with additional phenotypes reminiscent of defects associated with TANGO1/cTAGE5 knockouts[16], including pooled/clotted blood, darkened expanded yolk, and blistered skin (Fig. 2A–C). Phenotypic changes appeared more frequently in larvae treated with P2 + P5 at 4 μM supporting a dose-dependent effect of the treatment (Fig. 2C).

TANGO1 and cTAGE5 have been well established as essential for collagen secretion in various cell types and in vivo, including in zebrafish. To evaluate changes in extracellular collagen architecture after peptide treatment, we used second-harmonic generation (SHG) microscopy to visualize collagen fibers in the larval tail. Zebrafish were fixed, de-yolked, and imaged using SHG microscopy in the tail blade, a region with a high density of organized collagen fibers. Second harmonic images were acquired from three areas of the tail (highlighted squares in Fig. 2D, bright field). Images of collagen fibers from all three areas (Fig. 2D) suggested that control animals had a homogenous distribution in length and orientation, such that most fibers were well

aligned with respect to their neighboring fibers. On the other hand, fibers in peptide-treated tails were considerably more heterogeneous in size and their relative alignment (quantified as "dispersion" in Fig. 2E).

The altered ECM and changes in tissue architecture in peptide-treated zebrafish larvae are consistent with an inhibition of cTAGE5 and TANGO1 in vivo.

## Peptides bring about collagen retention in the ER

If the peptides inhibit TANGO1 or cTAGE5, they should reduce collagen export from the ER. The peptides were tested in two cell lines U2OS cells (collagen I) and RDEB/FB/C7 fibroblasts (collagen VII). Cells were incubated in a serum-free medium containing ascorbic acid (250 µM) and sodium ascorbate (1 mM) for 20 h to promote procollagen export from the ER.

Control U2OS cells or cells treated with peptide inhibitors (P2, P5, and P2 + P5 at 100 µM each) were fixed and processed for immuno-fluorescence microscopy to visualize intracellular collagen (Supplementary Fig. 2A). We observed increased collagen I (green) in peptide-treated cells, which colocalized with the ER-resident protein Calnexin (magenta), consistent with the peptides inhibiting TANGO1-dependent ER export of collagens. In RDEB/FB/C7 cells also (Supplementary Fig. 2B), peptide incubation resulted in a retention of collagen VII (green) in the ER as marked by Calnexin (magenta).

RDEB/FB/C7 cells treated with peptides show TANGO1 recruitment to accumulations of collagen. TANGO1 puncta are colocalized with ERES markers Sec31A (Supplementary Fig. 3), confirming that peptide inhibitors do not affect TANGO1 localization to ERES or its recruitment of collagen.

## Peptides inhibit collagen secretion in primary fibroblasts

To test if the peptides efficiently reduce collagen secretion also in primary cells derived from human individuals, we treated normal human skin fibroblasts, the key producers of collagen in connective tissues, with P2 and P5 and with the scrambled controls, scrP2 and scrP5 (composed of the same amino acids as P2 and P5, but in a different order). Fluorescently labeled peptides P2 and P5 entered cells readily, either individually or in combination, localizing throughout the cells (Fig. 3A, B). Only a slight intracellular accumulation of procollagen I was noted, which, however, did not reach statistical significance (Fig. 3C, D). Immunofluorescence microscopy of fibroblasts stained for collagen I showed a significant but mild accumulation of intracellular collagen after treatment with P2 and P5, (Supp. Figure 4). The addition of P2 or P5 alone did not alter collagen I secretion significantly (Fig. 3E, F). However, the addition of P2 + P5 resulted in a more than 60% reduction in secreted levels of procollagen I (Fig. 3G, quantified in H and I). We next asked if the effect exerted by P2 + P5 on primary human skin fibroblasts is reversible, and how long it lasts. Fibroblasts were first treated (or not, serving as controls) with P2 + P5 as earlier, and then supplied with growth-medium containing serum for 24, 48, or 72 h in the absence of peptides. Supernatants showed a strong reduction in collagen I levels by P2 + P5 after 24 h (Fig. 3J). Levels of secreted collagen I after 48 h were only mildly lower than the untreated controls, and after 72 h, the reduction was no longer detectable. These results indicate that P2 + P5 inhibits collagen secretion for at least 24 h after their removal and then their effect declines. Scrambled control peptides did not reveal any effect on collagen secretion at concentrations up to 40 µM (Fig. 3K). None of the peptides (P2, P5, P2 + P5 and scrP2+scrP5) exerted significant cytotoxic effects as determined by LDH release cytotoxicity assays (Fig. 3L) and cells showed similar morphology (Supplementary Fig. 5) and proliferation rates. Taken together, the data suggest that peptide treatment effectively inhibits collagen secretion. In addition, they also suggest that primary cells can clear most accumulated intracellular collagen better than the established cell lines.

## TANGO1/cTAGE5 inhibition reduces the secretion of multiple ECM proteins

Given that fibroblasts produce several types of collagen, including collagen I, III, and V, all of which contribute to the formation of heterotypic collagen I/III/V fibrils present in the large fibrillar network that characterizes the dermal ECM[32], we analyzed the supernatants from control and P2 + P5-treated fibroblasts. Mass spectrometric quantification of secretory cargoes (the secretome) confirms the significant reduction of the α1 and α2 chains of collagen I (COL1A1 and COL1A2), replicating our previous immunoblot results. We observed a significant reduction in secreted collagen III (COL3A1), collagen V (COL5A1), collagen XII (COL12A1), the latter often found decorating the surface of collagen I/III/V fibrils, and collagen VI (COL6A2 and COL6A3), the structural component of pericellular beaded filaments[32] (Fig. 4A and Supplementary Fig. 6). Interestingly, interfering with TANGO1 activity also impaired the secretion of some non-collagen ECM proteins that were previously not known to be TANGO1-cargoes, such as the core protein of the proteoglycan versican (VCAN), fibrillin1 (FBN1), a structural component of fibrillin-microfibrils[33], fibronectin, or laminin subunit α4 (LAMA4), a component of basement membranes[34]. There were no detected ECM proteins that were hypersecreted after treatment with P2 + P5. Overall, there were fewer hypersecreted proteins (depicted by orange symbols in Fig. 4A) than hypo-secreted (blue symbols) (Supplementary. Table 1). The highest apparent increase was MIA3 (TANGO1) itself, which is merely detection of the added peptides derived from TANGO1. Gene ontology (GO) analysis showed that the proteins primarily affected by peptide treatment include ECM structural components, particularly collagen (Fig. 4B). Several cargoes showed no significant change in secretion after treating cells with the peptides (Supplementary. Table 2), highlighting the specificity/selectivity of peptide activity.

## Controlled inhibition of wound healing in zebrafish

An important test of the inhibitors is whether they show efficacy in controlling collagen deposition into the granulation tissue during the wound healing process in vivo. We therefore studied the effect of the peptides on the healing of laser-induced cutaneous full-thickness wounds in adult zebrafish. In such wounds, granulation tissue is formed starting at two days post-wounding (2 dpw) and reaching maximal sizes at 4 dpw[29]. Fish wounds were microinjected both at 2 and at 3 dpw with PBS (control) or peptide (4 µM P2 + P5 in PBS). At 4 dpw, histology (Fig. 5A, B) and anti-type I collagen immunofluorescence labeling with DAPI-counterstaining of cell nuclei (Fig. 5C, G) revealed that peptide treatment resulted in reduced granulation tissue formation and reduced collagen deposition within the granulation tissue, while the density of cells/fibroblasts was slightly increased. In addition, TUNEL staining of dying cells did not reveal differences between controls and peptide-treated samples (Fig. 5H, I).

## Peptides inhibit collagen secretion in activated healthy and in patient-derived primary fibroblasts

We next aimed to assess if the identified peptides can reduce pathologically increased collagen secretion, such as that seen in activated fibroblasts and fibroblasts derived from patients with scleroderma. Activation was accomplished by treatment of fibroblasts with TGFβ, which prominently enhances collagen I production. Incubating TGFβ-activated skin fibroblasts from healthy individuals with P2 + P5 showed a clear inhibition of collagen secretion by the combination of P2 + P5 (Fig. 6A, left panel and B). The inhibitory effects were also clearly visible in scleroderma fibroblasts, especially after TGFβ activation (Fig. 6A, right panel and B). Most promisingly, both healthy and scleroderma fibroblasts showed reduced secretion of other abundant fibrotic ECM components, fibrillin1 and fibronectin (Fig. 6A, middle panels and B), further underscoring the importance of TANGO1 in

 

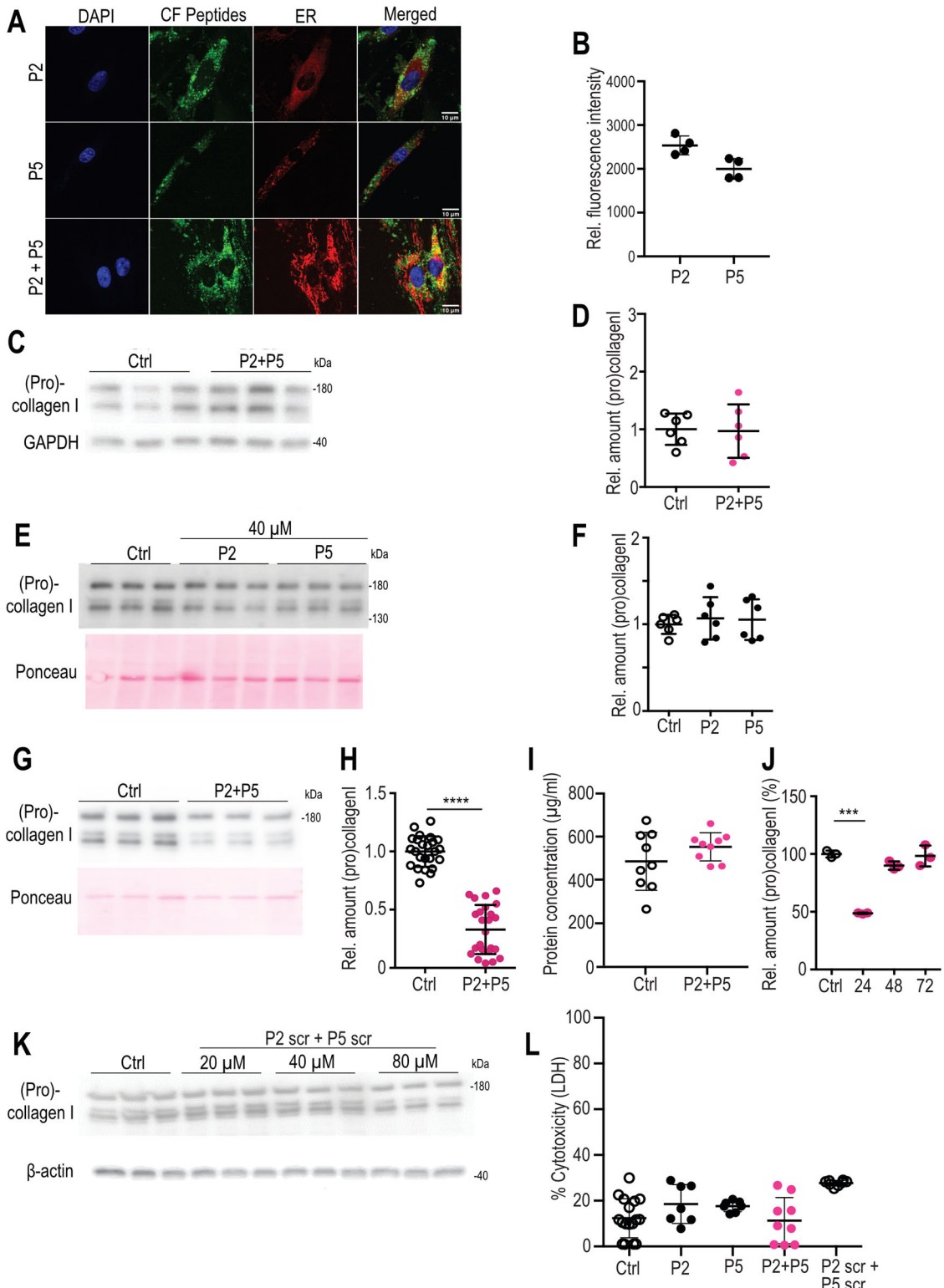

controlling pathophysiological secretion of bulky ECM secretory cargoes.

## Discussion

Here we show that membrane-permeant peptides that inhibit TANGO1 and cTAGE5 function, control fibrotic ECM protein secretion in scarring and fibrosis, from zebrafish to primary human fibroblasts from scleroderma patients.

Despite considerable preclinical development of potential anti-fibrotic therapeutics, only a few have entered clinical validation, likely due to the various redundant and concomitant pathways involved in the development of fibrosis. There is an urgent need for anti-fibrotics that positively impact morbidity and mortality. The peptides we present, provide a proof-of-principle that inhibiting TANGO1 and cTAGE5 at ERES could yield valuable anti-fibrotic reagents.

**Fig. 3 | Uptake of fluorescently labeled peptides into primary human dermal fibroblasts and inhibition of procollagen I secretion. A** Confocal micrographs of primary human dermal fibroblasts showing uptake of P2 + P5 supplied at 40 μM over 20 h in serum-free conditions. Co-staining with an ER marker in red. Bar: 10 μm. **B** Quantification of uptake of fluorescent peptides (*N* = 4, mean +/− SD of intracellular fluorescent intensity). **C, D** Representative western blot (**C**) and quantification of fibroblast lysates treated or not (Ctrl) with P2 + P5 at 40 μM for 20 h. *N* = 6 (**D**). **E, F** Representative western blot from supernatants; Ponceau staining reflects the loading control in the corresponding cell lysates (**E**). Quantification of supernatants of human fibroblasts treated for 20 h in serum-free conditions with either P2 or P5 at the indicated concentration. *N* = 6 (**F**).
**G, H** Representative western blot from supernatants; Ponceau staining reflects the loading control in the corresponding cell lysates (**G**). Quantification of supernatants from untreated and P2 + P5 treated skin fibroblasts. *N* = 24; Mann– Whitney *U* test, two-tailed, **** *p* < 0.0001 (**H**). **I** Protein concentration of untreated and P2 + P5 treated cell lysates. N = 9. **J** Duration, in hours, of inhibitory effect after removal of P2 + P5. Levels of untreated (Ctrl) were set at 100%. *N* = 3 per time point, two-way ANOVA with Sidak's multiple comparisons test. *** *p* < 0.0001. **K** Collagen I western blot of supernatants from controls and scrambled (scr) peptide-treated control fibroblasts and β actin western blot from corresponding cell lysates.
**L** Determination of cytotoxicity of control fibroblasts treated with different peptides (measured by lactate dehydrogenase release (**L, D, H**)). *N* = 17 (Ctrl)/7 (P2)/7 (P5)/9 (P2 + P5)/8 (P2 scr + P5 scr). *N* = independent samples. Data shown as mean ± SD.

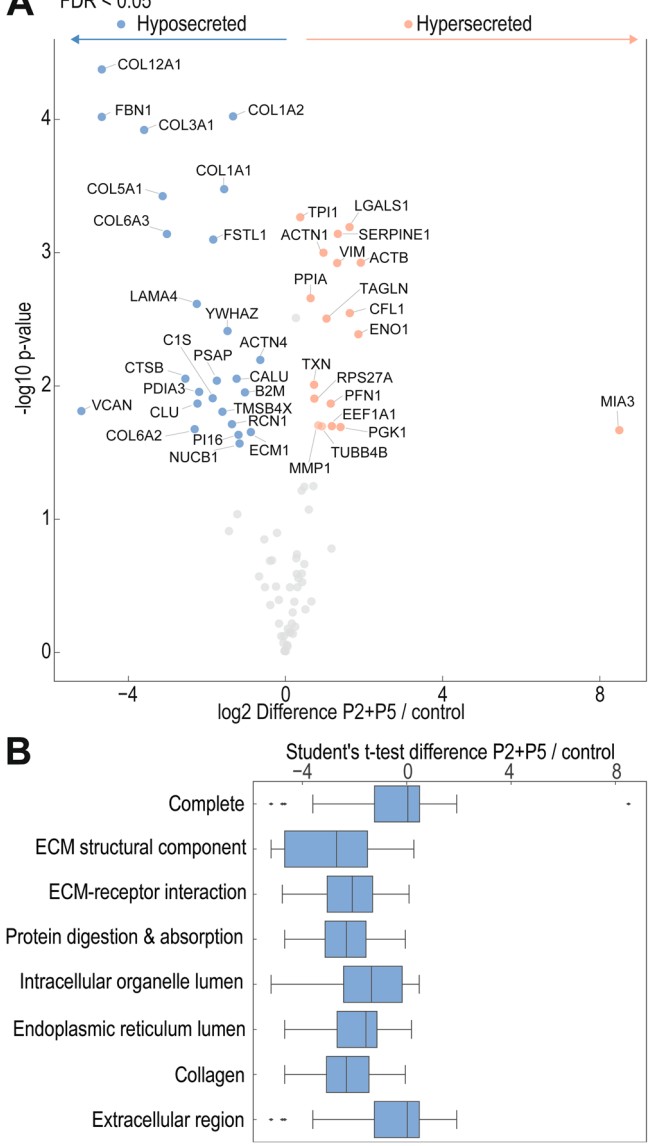

**Fig. 4 | Effects of peptide treatment on the secretome of human fibroblasts. A** Volcano plot illustrating significantly downregulated (blue symbols) and upregulated (orange symbols) proteins, by treatment of primary human dermal fibroblasts with 40 μM P2 + P5 for 20 h in serum-free conditions in comparison to control samples. Unpaired two-sided Student's t-test, $S_0$ = 0.1, *N* = 3 independent samples. **B** Gene ontology (GO) terms of proteins affected by peptide treatment. Box plot represents the median, 25th and 75th percentiles, max. and min. are connected through whiskers. Outliers are defined as Q1−1.8 interquartile range (IQR) and Q3 + 1.8 IQR. An unpaired two-sided Student's test was performed.

TANGO1 and cTAGE5 bind each other constitutively[10,35] and depleting TANGO1 results in a concomitant loss of cTAGE5[36]. Global, quantitative mapping of protein copy numbers in mammalian cells revealed that there are ~173,000 molecules of TANGO1 and cTAGE5 each per cell[37]. At an ERES, TANGO1 and cTAGE5 have complementary functions. While TANGO1 recruits export-competent collagen in the ER lumen, TANGO1, and cTAGE5 together build an export route by scaffolding several cytoplasmic interactions, including Sec12 recruitment by cTAGE5[10,21,38,39]. TANGO1 and cTAGE5 interact robustly and could be considered a single unit, raising the possibility that their dimeric state is more stable. We speculate that this 1:1 stoichiometric balance of protein abundance is maintained by the stabilization of a TANGO1–cTAGE5 complex, mediated by their binding interface of interacting coiled-coil regions. Individual proteins could be unstable unless in complex with their binding partner. It is possible that peptide treatment inhibits TANGO1-cTAGE5 heterodimerization, thereby decreasing their stability. In such a case, peptide treatment would ensure that an antifibrotic effect could continue even after peptides are cleared from inside the cell, at least until TANGO1 and cTAGE5 levels have been replenished. Measurement of the interaction and any loss thereof after peptide treatment was not possible due to technical reasons, but the observed effect on protein levels is consistent with such a disruption of heterodimerization.

The zebrafish model has proven a useful in vivo system for a range of biological mechanisms from tissue development to wound healing, and drug discovery. By testing the peptides in zebrafish larvae, we confirmed that they recapitulate published phenotypes observed in a TANGO1 knockout animals[16]. We found that the inhibitors affect larval size, consistent with observations from TANGO1 or cTAGE5 mutant fish[16], a global knockout of TANGO1 in mouse embryos, or conditional knockouts of cTAGE5 in mice[12,40,41]. An enlarged and darker yolk in treated larvae is indicative of disruptions in lipoprotein trafficking into the zebrafish embryo during development, consistent with previous studies[42–44]. Visualization of ECM architecture in the tail revealed a change in collagen fiber deposition and structural organization, in line with the central role of TANGO1/cTAGE5 in collagen secretion. In addition, we used zebrafish as a model to study collagen deposition in the newly forming tissue after laser-induced wounding. Inhibiting TANGO1/cTAGE5 by the peptides slowed granulation tissue formation without demonstrating evidence for toxicity. These studies demonstrate that modulating TANGO1/cTAGE5 activity is a valuable tool to dissect and understand the complex processes required for proper tissue repair and at the same time might also provide a novel therapeutic approach to interfere with the pathophysiological mechanisms driving excessive ECM deposition in aberrant wound healing and fibrosis including hypertrophic scarring. Skin as a target organ in scleroderma and other sclerosing skin diseases offers to use peptide application as local injections infiltrating the involved tissue. In some situations, e.g. to reduce excessive scar formation following injury even topical applications allowing direct penetration of the dermal compartment might reduce potential side effects.

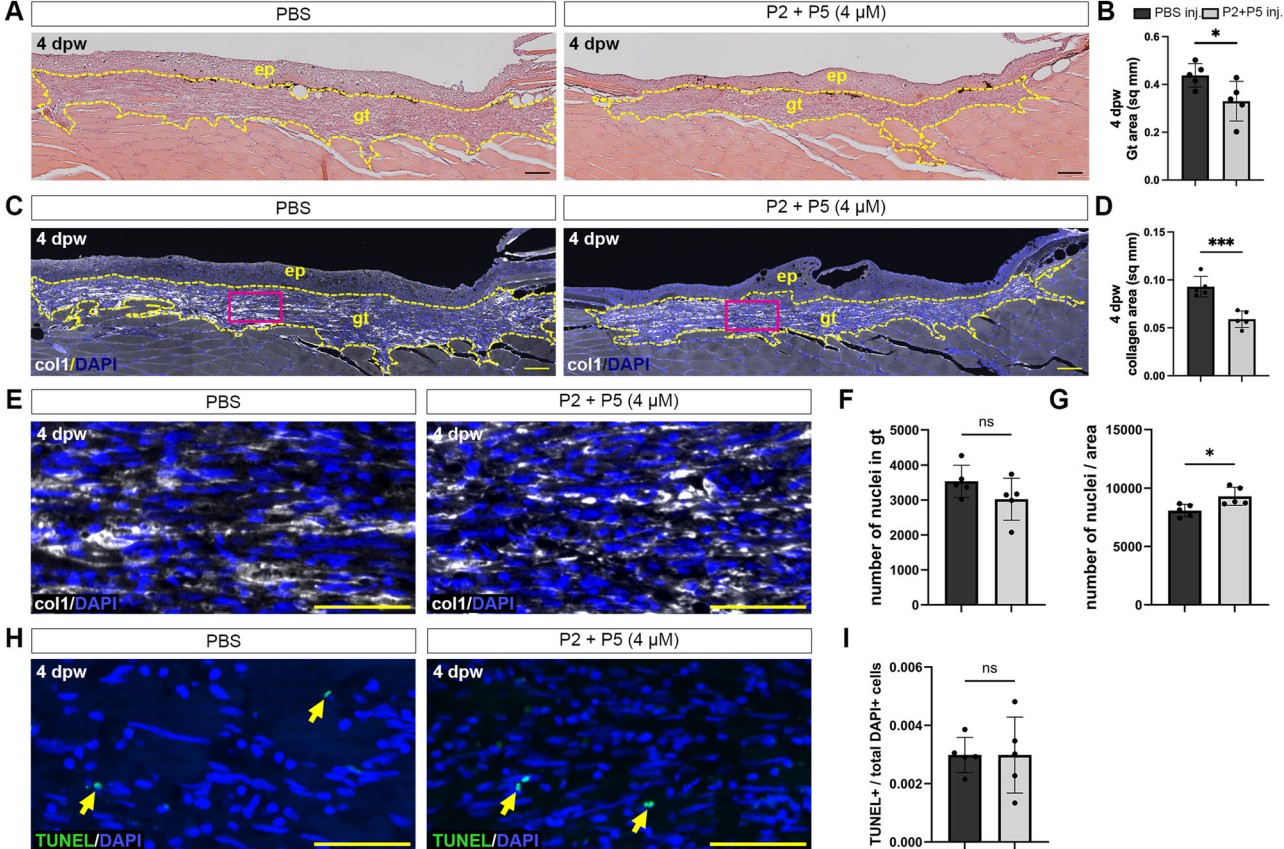

**Fig. 5 | Peptides limit the formation of granulation tissue in zebrafish wounds.**
**A** H&E staining of skin longitudinal sections at 4 dpw from control PBS-treated fish (left), or those treated with P2 + P5 (right). Granulation tissue (gt) area was measured (yellow dashed lines). **B** Quantification of granulation tissue area in control or peptide-treated animals. * $p = 0.0369$. **C** Immunohistochemical analysis of collagen deposition during wound healing in the granulation tissue in control (left) or peptide-treated (right) animals. **D** Quantification of collagen area in the granulation tissue. *** $p = 0.0006$. **E** Higher magnification of immunohistochemical analysis of collagen deposition and nuclei (magenta boxes in **C**). **F** Quantification of total nuclei within the granulation tissue. **G** Quantification of total nuclei divided by the granulation tissue area. * $p = 0.0198$. **H** TUNEL staining of 4 dpw wounds in granulation tissue in control (left) or peptide-treated (right) animals. **I** Quantification of TUNEL-positive cells divided by total DAPI-positive cells in the granulation tissue. Scale bars: **A**, **C** = 200 μm; **E**, **H** = 50 μm. ep: epidermis. $N = 5$ fish per treatment, Student's *t*-test, two-tailed. Non-significant (NS). Data are shown as mean ± SD.

Fibroblasts during wound healing and in fibrotic processes are activated by profibrogenic cytokines and growth factors, which are released from inflammatory and/or endothelial cells[5]. These activated fibroblasts have the characteristics of myofibroblasts and are involved in tissue repair and scar formation. The potent profibrotic cytokine, TGFβ, has been shown to be crucial for these processes and plays a major role in myofibroblast formation. In human fibroblasts from healthy individuals, TGFβ induces collagen synthesis and increased secretion into the medium. It is promising that the peptide inhibitors were particularly effective at reducing the secretion of multiple ECM proteins in primary fibroblasts and TGFβ-activated control fibroblasts (Figs. 4 and 6) without exerting cytotoxic activity. Myofibroblasts in scleroderma likely remain in an activated state, causing an uncontrolled deposition of ECM proteins and assembling a disturbed ECM, characteristic of a rigid and fibrotic tissue structure. In vitro, these cells are particularly responsive to TGFβ treatment (Fig. 6). Inhibiting TANGO1/cTAGE5 resulted in a severe reduction of collagen secretion also in these cells suggesting that myofibroblasts in patients with scleroderma can be targeted by the inhibitory peptides.

TANGO1/cTAGE5 inhibition would arrest collagens in the ER, from where they are likely cleared by a combination of autophagy, ER-phagy and proteasomal degradation, leading to our observation that human fibroblasts showed only a minor accumulation in intracellular collagen when secretion was inhibited.

In addition to the potential implications for anti-fibrotic therapeutic development, it is exciting that these peptide inhibitors were active in zebrafish, as they provide an experimental handle to control ECM secretion selectively, acutely, and reversibly, in almost any metazoan system without the confounds of cells or tissues adapting to the genetic loss of TANGO1. Peptides will only act in a localized manner as they can only penetrate the outermost layer of cells that they are exposed to.

We see a clear role for TANGO1 in secreting several large ECM proteins including many collagens, fibrillin 1, and fibronectin, among others. Our data support the idea that ECM proteins are the secretory cargoes most sensitive to perturbation of TANGO1 function. This could be an indication of the enormous secretory load during matrix secretion, as TANGO1 family proteins are likely required for the ER export of bulky and/or cargoes secreted at high volume. This effect could arise from how TANGO1 family proteins (re)organize or scaffold the early secretory pathway to accommodate these cargoes. An additional intriguing possibility that merits further study, is that several of these ECM proteins and/or their modifying enzymes exit the ER in a concerted manner, perhaps already interacting with each other in a rudimentary 'mini-ECM.'

The exact mechanism remains incompletely understood, and a detailed investigation is crucial for a better understanding of the tightly balanced and controlled ECM formation and its organization in development and tissue homeostasis.

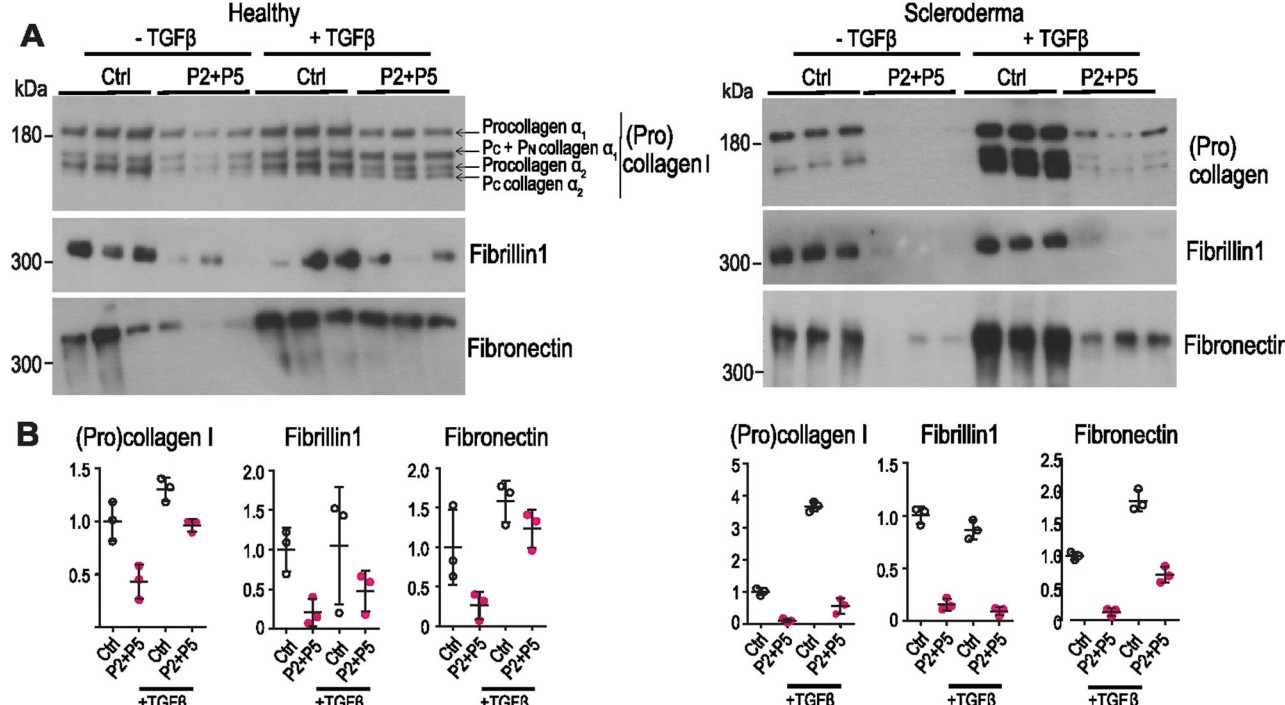

**Fig. 6 | Peptides inhibit ECM protein secretion by TGFβ-activated healthy and scleroderma fibroblasts. A** Western blot of supernatants from fibroblasts incubated for 20 h in the absence of serum without (Ctrl) or with 40 µM each of P2 + P5 and without or with 10 ng/ml TGFβ. Peptide treatment strongly reduced amounts of secreted (pro)collagen I (top), fibrillin1 (mid) and fibronectin (bottom) in fibroblasts from healthy donors and scleroderma patients before and particularly after stimulation of ECM production by TGFβ. **B** Quantification of (**A**). $N = 3$ independent samples. Data are shown as mean ± SD.

## Methods

### Peptide synthesis

Peptides were synthesized by automated solid-phase peptide synthesis (SPPS) (peptide synthesizer from MultiSynTech, Syro I) using the Fmoc (fluorenylmethoxycarbonyl)/t-Bu (tert-butyl) strategy on Rinkamid resin (loading 0.48 mmol, 0.015 mmol scale). Amino acids were coupled using 8 equivalents (eq) N,N′-diisopropylcarbodiimide (DIC) and ethyl cyanohydroxyiminoacetate (Oxyma) in dimethylformamide (DMF) as activating reagents. Fmoc was cleaved off using 20% piperidine in DMF. Finally, the resin was washed with dichloromethane (DCM), methanol (MeOH) and diethyl ether (Et$_2$O), respectively.

For cell uptake and fluorescence microscopy studies, also 5,6-carboxyfluorescein (CF)-labeled peptides were synthesized. Therefore, 5 eq CF, Oxyma and DIC were applied and peptides were labeled at the N-terminal while still attached to the resin.

After complete synthesis, peptides were cleaved by treating the resin for 3 h at room temperature with a mixture of concentrated (conc.) trifluoroacetic acid (TFA)/triisopropylsilane (TIS)/H2O (95:2.5:2.5, v/v/v). Subsequently, peptides were precipitated in ice-cold diethyl ether. Reversed-phase (RP) RP-high pressure liquid chromatography (HPLC) (column: CC 125/4.6 Nucleodur 100-5 C18e (Macherey-Nagel) using a gradient from 10–60 % of acetonitrile (ACN) in H$_2$O with 0.1% formic acid for sample separation, followed by electrospray ionization-mass spectrometry (ESI-MS, Thermo Scientific LTQ-XL) was used for identification. Purification was achieved by semi-preparative RP-HPLC (Hitachi Elite LaChrom) on a VP 250/8 Nucleodur 100-5 C18e column (Macherey-Nagel) using linear gradients from 10–60% B in A (A = 0.1% conc. TFA in water; B = 0.1% conc. TFA in ACN) over 45 min and a flow rate of 1.5 ml/min. Final purity of all compounds was >50 %. Finally, to exchange the remaining TFA by chloride peptides were diluted in 3 mL 80 mM HCl solution, incubated for 2 h shaking, and lyophilized overnight. This procedure was repeated two times.

### Imaging intracellular collagen in cell culture

Collagen visualization was carried out as before[9,21]. Briefly, RDEB/FB/C7 cells grown in six-well plates were washed with PBS to remove the medium and replaced in fresh DMEM supplemented with 10% FCS. The cell permeable peptides (P2, P5) were then added at 10 µµM final concentration for 20 h, with ascorbate (250 µM ascorbic acid, 1 mM phospho-ascorbate) in full medium.

U2OS cells grown in petri dishes were washed to remove the medium and replaced in fresh medium. The cell permeable peptides (P2, P5) were then added at 10 µM final concentration for 4 h, with ascorbate (250 µM ascorbic acid, 1 mM phospho-ascorbate) in Opti-MEM. Control has nothing added.

Cells were fixed and processed for immunofluorescence microscopy.

### Zebrafish experiments

**Zebrafish maintenance.** For the peptide incubation experiments, AB wild-type zebrafish were used. Zebrafish were maintained at Parc de Recerca Biomèdica de Barcelona (PRBB) according to the standard procedures of the aquatic facility. All the protocols followed the national and European guidelines and were approved by the Institutional Animal Care and Use Ethic Committee (PRBB–IACUEC). The principles of the 3Rs were applied in all zebrafish experiments.

The zebrafish wound healing experiments were approved by the local and federal animal care committees (City of Cologne: 8.87-50.10.31.08.129; LANUV Nordrhein-Westfalen: 81-02.04.2018.A097).

**Incubation.** Zebrafish embryos were kept in E3 medium (5 mM NaCl, 0.17 mM KCl, 0.33 mM CaCl$_2$, 0.33 mM MgSO$_4$) prior to the experiment. Zebrafish embryos were manually dechorionated between 2– and 4–cell stages to ensure that all eggs used in the experiment were viable. A total of 20 embryos per condition were immediately transferred to a four-well plate with 5 embryos per well (Nunc delta, Thermo

Fisher 176740) pre-filled with 500 μl of E3 medium with peptide inhibitors (2 μM or 4 μM of both P2 and P5). Control embryos were kept in E3 only, and medium changes were carried out at 1 dpf and 3 dpf for all treatment conditions. Embryos and larvae were continuously treated over the duration of the experiment and kept at 28°C. To avoid pigmentation in the zebrafish larvae, 0.1% PTU was added to the medium from 24 hpf. Embryos were monitored at sphere, shield, 75% epiboly, 1, 2, and 3 dpf for survival and development. At 3dpf, a representative group of larvae were imaged for phenotyping and length measurements. The experiment was carried out three times. For imaging of collagen fibers with second harmonic generation microscopy, larvae were fixed in 4% PFA at 4 °C overnight, followed by a PBS wash.

### Wound healing in zebrafish

Adult zebrafish of the TL/Ekwill wildtype strain at an age of 8 months were used for wounding experiments. Full-thickness wounds of adult fish with a diameter of ~2 mm were introduced with a Dermablate laser, as described previously[29]. In short, adult fish were anesthetized in 0.13% Tricaine (w/v) and placed in a lateral position on Whatman paper soaked in the system water. Full-thickness wounds were introduced on the left flank directly anterior to anal and dorsal fins. An Erbium:YAG MCL29 Dermablate dermatology laser (Asclepion, Jena, Germany) was set to a frequency of 1 Hz and 5 pulses with a strength of 500 mJ were applied.

For local treatment of wounds with peptides, wounds were microinjected both at 2 and at 3 days post wounding (dpw), using a glass needle with a tip diameter of less than 50 μm, a micromanipulator and a Pneumatic PicoPump (PV820, World Precision Instruments). Wounds were injected at five sites, an inner site directly in the center of the wound and four outer sites approximately 0.3 mm from the edge of the wound and at maximal distances from each other, thus forming a pattern as the five dots on a game die. Per wound, a total of 80 nl of a 50 μM peptide stock solution in phosphate buffered saline (PBS), pH 7.4 (Sigma) was injected. Assuming a granulation tissue volume of ~1 mm³ (μl), as calculated from its dimensions in our histological sections, this should yield a final peptide concentration within the granulation tissue of ~4 μM. As control, plain PBS was injected. For histological analyses, wounded and injected fish were sacrificed at 4 dpw and fixed in 4% paraformaldehyde/ PBS overnight at room temperature, decalcified in 0.5 M EDTA (pH 7.4) at room temperature for 5 days, dehydrated in a graded series of alcohols, cleared in Roti-Histol (Carl Roth, Karlsruhe, Germany) and embedded in paraffin. Sections of 10 μm thickness were stained with hematoxylin and eosin using standard protocols. For quantification of granulation tissue sizes, consecutive sections of wounds were stained with hematoxylin and eosin. Sections comprising the centre of the wound were selected, granulation tissue areas were measured using ImageJ, and Student's t-test was used to calculate significance.

For immunohistochemical analysis, consecutive paraffin sections were microwaved in a preheated 10 mM citrate buffer (pH 6.0, 0.05% Tween 20) for 15 min following rehydration. The slides were then left to cool at room temperature for 20 min and washed with PBS-Tween 20 (0.1%) three times for 10 min each. Blocking (10% FBS + 1% DMSO in PBST) was done for 3 h at room temperature. Anti-col1 (1:200, ab23730, Abcam, Cambridge, MA, USA) primary antibody in blocking buffer was added to the slides for incubation overnight at 4 °C. The following day, the slides were washed with PBST four times for 10 min each and goat anti-rabbit Cy3 (1:750, RRIDAB AB_2534029, cat. no. A10520) secondary antibody in the blocking buffer was added to the slides for incubation at room temperature for 90 min. The slides were then washed again with PBST four times for 10 min each and mounted with Mowiol including DAPI. The slides were stored at 4 °C until further imaging. Images were captured on a Zeiss Apotome microscope and the collagen-I-stained area in the granulation tissue were measured using ImageJ and Student's t-test was used to calculate the significance.

The ECM production was analyzed at 4 dpw since the collagen accumulation in the granulation tissue of zebrafish cutaneous wounds reaches its maximum at this stage[29].

### TUNEL staining

TUNEL staining on paraffin sections of the same individuals, also used for the Col2 immunofluorescence analysis, was performed using the DeadEndTM Fluorometric TUNEL System (Promega, G3250) and following the manufacturer's protocol, except prolonging the Proteinase K (provided in the kit) treatment to 30 min and adding 2 PBS-TritonTM-X-100 0,1% wash steps following the rTdT enzyme reaction. Specificity of the assay was confirmed via negative and positive (DNAse treatment) controls. Images were taken on a Zeiss Axio Imager Z.1 microscope with an apotome function. Images were processed using the mpl-inferno LUT application of the ImageJ software (Version 1.54 f). For quantification, five individual fish and two individual sections per fish were analyzed.

### Second Harmonic Generation (SHG) microscopy of Zebrafish fin collagen fibers

To visualize collagen fibers, the yolk of the fixed larvae was removed manually to improve imaging. PFA-fixed and de-yolked zebrafish were mounted in 1% low melting point agarose (Invitrogen 16520050) on 50 mm bottom-glass dishes (P50G-0-14-F, MatTek Corporation) sandwiched with a round cover glass (30 mm diameter #1, Menzel Gläser) and sealed with nail polish, to allow imaging in forward and backward direction. The experiment was carried out three times and a total of three control and three peptide-treated fish were imaged using second harmonic generation microscopy. The optical setup consists of a confocal microscope (Eclipse Ti-U, Nikon) adapted to perform nonlinear microscopy[45]. The system was pumped with a femtosecond-pulsed laser beam (Halite, Fluence) operating at a central wavelength of 1030 nm. The laser was transversally scanned with a pair of two galvanometric mirrors in tandem configuration (Cambridge Technology). The beam was expanded with a telescope arrangement to fill the microscope objective pupil and directed with a dichroic mirror (DMS1000R, Thorlabs) to a 20x NA 0.75 air objective (Plan Apo λ, Nikon) to focus the beam on the sample. The signal was detected in forward direction with a 25x NA 1.10 water objective (Apo LWD, Nikon). The SHG signal is collected with a photomultiplier tube (H7422-40, Hamamatsu) using a dichroic mirror (FF665-Di01), an IR filter (FGB37-A, Thorlabs) and a bandpass filter centered at 515 nm (FF01-515/30-25, Semrock). The transmitted light from the femtosecond-pulsed laser was collected with a silicon photodiode for sample reference. Image acquisition was obtained with an in-house implemented LabVIEW-based software. Z-stacks of 512 × 512 pixel images were reconstructed with FIJI (ImageJ, version 1.50i).

We quantified the preferred orientation of the collagen fibers from Control and P2 + 5 (4 μM)-treated fishes using the Directionality plugin from FIJI (ImageJ, version 1.50i). Data normalization and plotting were performed using GraphPad Prism version 9.3.1 for Windows, GraphPad Software.

### Fibroblasts from human donors

Skin biopsies were obtained from female and male healthy individuals or patients with scleroderma (systemic sclerosis) after written consent and with approval of the local ethics committee (Ethics committee of the medical faculty of University of Cologne, 21-1072). The study was conducted in accordance with the criteria set by the Declaration of Helsinki. Primary dermal fibroblasts were cultured by explant outgrowth in DMEM (Gibco Invitrogen, 41965-039) with 100 U/ml penicillin, 100 μg/ml streptomycin (Sigma, P0781), 2 mM L-glutamine (Biozym, 882027) 50 μg/ml Na-ascorbate (Sigma-Aldrich, A4034) and 10% fetal calf serum (Gibco, 10270-106) in the moist atmosphere of a $CO_2$ incubator[46]. Cells were used in passages 2–5.

## Peptide uptake into primary skin fibroblasts

For visualization of peptide uptake, fibroblasts were seeded at $1.4 \times 10^4$ into an eight-well-chambered coverslip (Ibidi) and grown confluent overnight. To visualize the ER, cells were treated with CellLight™ ER-RFP (Invitrogen) for 24 h. Then, peptide solutions of serum-free medium containing either 40 μM CF-labelled P2 or P5, or both peptides combined were added. After 20 h, cell nuclei were stained with Hoechst 33,342 nuclear dye (Thermo Fisher) for 10 min. Live fluorescence microscopy images were taken using the Keyence BZ-X800 microscope and afterwards, images were processed using ImageJ.

## Secretion assays in fibroblasts from human donors

To assess secretion, fibroblasts were seeded at high density ($3 \times 10^5$ per 6-well) and grown to confluence overnight, washed extensively and treated in serum-free medium with either P2 or P5 or the combination of both peptides or the scrambled peptides scrP2, scrP5 or the combination of them at the indicated concentrations for 20 h. For some experiments, fibroblasts were pretreated with TGFβ1 (10 ng/ml; R&D systems, 240-B) for 12 h in the presence of 10% fetal calf serum, washed extensively and serum-free media with TGFβ1 and peptides were added for 20 h.

To assay the duration of the effect of the peptides, cells were treated without (Ctrl) or with 40 μM each of P2 + P5 for 20 h in the absence of serum, and supernatants were removed. Then, growth medium containing 10% serum but no peptides was supplied to control and peptide-treated cultures for 24, 48 or 72 h.

Equal volumes of supernatants were analyzed by SDS-PAGE and transferred to PVDF membranes (Immobilon, Merck, 0.45 μm, IPVH00010). This was possible because total protein concentrations in the corresponding lysates were comparable, as secured by BCA determination of total protein content (Pierce™ BCA Protein Assay Kits (Thermo Scientific, 23225)) and Ponceau staining (Sigma-Aldrich, P7170) of membranes with proteins from the cell lysates. Primary antibodies used were: Goat anti-type I collagen (SouthernBiotech, 1310-01; 1:2000; reduced); rabbit anti-fibrillin1 (kind gift from G. Sengle, Cologne; 1:2000 unreduced); rabbit anti-fibronectin (abcam, ab23750; 1:500, unreduced); and rabbit anti-GAPDH (abcam, ab9485; 1:2500; reduced), followed by appropriate horseradish peroxidase-conjugated secondary antibodies and detection using enhanced chemiluminescence (Thermo Scientific 34087). Intracellular collagen I retention was analyzed in fibroblast lysates (20 mM Tris-HCl pH 7.5, 1% SDS, 0.5% NP-40, 2 mM EDTA) and normalized to GAPDH (abcam, ab9485; 1:2500; reduced). Band intensities were measured using the Gel Analysis tool in ImageJ software.

Cytotoxic effects induced by peptide treatment were determined by lactate dehydrogenase (LDH) levels in fibroblast supernatants using Cytotoxicity Assay (Promega G1780) according to the manufacturer's instructions.

## Secretome analysis

Sample preparation. Supernatants from treated fibroblasts and controls were also used for proteomics analysis to determine the secretome. Supernatants were removed from the cultures, filtered and clarified through centrifugation. A protease inhibitor was added, and samples were frozen at − 80 °C until further processing. Proteins from supernatants were acetone precipitated and resuspended in 6 M urea/2 M thiourea. Disulfide reduction and alkylation were performed using 20 mM TCEP (Sigma-Aldrich) and 25 mM CAA (Merck) at RT for 1 h. Proteins were digested with endoproteinase Lys-C (Wako) (1:100 enzyme-to-substrate ratio) for 3 h at RT. Then, the urea concentration was diluted fourfold with 50 mM ABC buffer and digested with trypsin (Promega/Sigma) (1:100 enzyme-to-substrate ratio) overnight at RT. Digestion was stopped by adding formic acid at a final concentration of 1%. The samples were desalted by Stop-and-Go extraction using SDB-RPS StageTips[47].

LC-MS/MS analysis and data processing. Secretome analyses were performed using an Easy nLC 1000 ultra-high performance liquid chromatography (UHPLC) coupled to a QExactive Plus mass spectrometer (Thermo Fisher Science) with the same settings as described before[48]. Acquired MS spectra were correlated to the human FASTA database using MaxQuant (V. 1.5.38) with its implemented Andromeda search engine[49]. All parameters were set to default. N-terminal acetylation and methionine oxidation were set as variable modifications and cysteine carbamidomethylation was set as fixed modification. Statistical analyses and GO annotations were performed in Perseus (V. 1.6.2.3.58) and data were visualized using InstantClue[50]. The significance cutoff was set to an FDR < 0.05.

## Statistical analyses

Statistical analyses were done using GraphPad Prism version 6. Applied tests are indicated in individual figure legends. * $= p < 0.05$; ** $p < 0.01$; *** $p < 0.001$; **** $p < 0.0001$.

## Reporting summary

Further information on research design is available in the Nature Portfolio Reporting Summary linked to this article.

## Data availability

The mass spectrometry proteomics data have been deposited to the ProteomeXchange Consortium via the PRIDE partner repository[51] with the dataset identifier PXD041678. All unique materials generated are readily available from the authors. Source data are provided in this paper.

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

## Acknowledgements

The authors thank the CRG Advanced Light Microscopy Unit (ALMU), the Protein Technologies Unit, and the CRG/UPF Flow Cytometry Unit. Susanne Neumann (Cologne) provided excellent technical assistance. We would like to thank Prof. Dr. Pia Moinzadeh for obtaining the skin biopsies. The authors acknowledge the support of the Spanish Ministry of Science and Innovation to the EMBL partnership, the Centro de Excelencia Severo Ochoa and the CERCA Programme/Generalitat de Catalunya and the Alexander v. Humboldt foundation. We acknowledge financial support from the following sources: Ministerio de Economía y Competitividad (SEV-2012-0208, BFU2013-44188-P, CSD2009-00016 to V.M.); Deutsche Forschungsgemeinschaft (ID 468236352) and CMMC Cologne to T.K and V.M. V.R. acknowledges financial support from the Ministerio de Ciencia y Innovacion through the Plan Nacional (PID2020-117011GB-I00). This project has received funding from the European Research Council (ERC) under the European Union's Horizon 2020

research and innovation programme (grant agreement No 951146). This publication is part of the Project TARTAFI (Ref. PDC2021-121870-I00), funded by MCIN/AEI/10.13039/501100011033 and the European Union "NextGenerationEU"/PRTR. This work was supported by the Fondation pour la Recherche Médicale, (FRM) AJE202210016216 to I.R. M.C, G.C, F.C, J.A, and PL-A acknowledge funding from Fundació CELLEX; Ministerio de Economıa y Competitividad - Severo Ochoa programme for Centres of Excellence in R&D (CEX2019-000910-S); CERCA programme and Laserlab-Europe (871124); Ministerio de Ciencia e Innovación (MCIN/AEI/10.13039/501100011033); and Fondo Social Europeo (PRE2020-095721, M.C). The SLN facility corresponds to a "Grup reconegut 2021 SGR 01456 pel Departament de Recerca i Universitats de la Generalitat de Catalunya". F.C. acknowledges support from the Government of Spain (RYC-2017–22227, PID2019-106232RB-I00/10.13039/501100011033/110198RB-I00).

## Author contributions

Conceptualization: I.R., V.M., T.K., I.N. and B.E. Experimentation: I.R., A.-H.R., I.K., M.S., L.K, P.F., K.H., J.N., A.B., H-MH., C.V., K.B., M.C., G.C, M.V, B.D., R.K.Z., and R.K. Data analysis: all authors. Drafting manuscript and approval: all authors. Resources and supervision: C.C., F.C., J.A., P.L.-A., V.R., C.J., M.K., M.H., B.E., I.N. and T.K. Funding acquisition: I.R., T.K., V.M., V.R., M.C., G.C., F.C., J.A. and P.L.-A.

## Competing interests

The authors declare no competing interests.

## Additional information

[1]Centre for Genomic Regulation (CRG), The Barcelona Institute of Science and Technology, Dr Aiguader 88, Barcelona, Spain. [2]Université Paris Cité, CNRS, Institut Jacques Monod, Paris, France. [3]Translational Matrix Biology, University of Cologne, Medical Faculty, Cologne, Germany. [4]European Molecular Biology Laboratory, EMBL Barcelona, Dr. Aiguader 88, PRBB Building, Barcelona, Spain. [5]Institute of Zoology, Developmental Biology, Biocenter Cologne, University of Cologne, Cologne, Germany. [6]Cologne Excellence Cluster on Cellular Stress Responses in Ageing-Associated Diseases (CECAD), University of Cologne, Cologne, Germany. [7]Department of Chemistry, Institute of Biochemistry, University of Cologne, Cologne, Germany. [8]ICFO-Institut de Ciencies Fotoniques, The Barcelona Institute of Science and Technology, Barcelona, Spain. [9]Max Planck Institute for Biology of Aging, Cologne, Germany. [10]IFOM-inStem Joint Research Laboratory, Centre for Inflammation and Tissue Homeostasis, Institute for Stem Cell Science and Regenerative Medicine (inStem), Bangalore, Karnataka, India. [11]Universitat Pompeu Fabra (UPF), Barcelona, Spain. [12]ICREA, Pg, Lluis Companys 23, Barcelona, Spain. [13]Center for Molecular Medicine (CMMC), University of Cologne, Cologne, Germany. ✉e-mail: ishier.raote@ijm.fr; ines.neundorf@uni-koeln.de; vivek.malhotra@crg.eu

