## [Peer Review File · Nature Communications]

TANGO1 inhibitors reduce collagen secretion and limit tissue scarringREVIEWER COMMENTS

Reviewer #1 (Remarks to the Author):

Raote et al generated membrane-permeable peptides that inhibit the interaction between TANGO1 and cTAGE5, resulting in the inhibition of collagen formation. This effect was validated in cultured fibroblasts from healthy individuals and from scleroderma patients. The peptide inhibitors also reduced granulation tissue formation during wound healing in zebrafish. These results suggest the use of the peptide inhibitors for the treatment of excessive scar formation and fibrosis. The role of TANGO1 and cTAGE5 in collagen secretion is not novel, but the development of the peptide is innovative and the translational relevance is strong. However, there are also some weaknesses of this study as detailed below.

1. The results shown in Fig. 1G are not convincing. There is obviously only an effect of the peptides at the highest concentration, and it is unclear if this reduction is statistically significant. Please include a statistical analysis.
2. The Western blots shown in Fig. 1H and 1I look convincing, but the quantitative analysis (Fig. 1J) is not convincing. There is obviously a high variability, and the differences are not statistically significant.
3. There is a general problem with the controls in the in vitro and in vivo assays. The authors used PBS or no treatment as control, but the appropriate control would be a membrane-permeable peptide, which does not inhibit TANGO1. It is well known that membrane-permeable peptides can have toxic effects. Such a control should be used – at least for the in vitro experiments.
4. Fig. S2: Why is there reduced survival in some of the controls? Why is there such a high variability between experiments when 4 μ M of the peptides are used? Overall, it is unclear how the data shown in this figure should be interpreted.
5. Fig. 2E: It is unclear if any of the differences are statistically significant.
6. Please quantify the data shown in Fig. S3.
7. Fig. S5: It seems that the peptides even inhibit toxicity – please comment on this result.
8. Fig. 3E and 3F: Why is the combination of both peptides required to inhibit collagen secretion? Is there also less collagen I in the deposited matrix?
9. The results shown in Fig. 4 are interesting, but there is no further validation of the data. The authors should select 3-4 regulated proteins and check their amounts in the secretome, in the deposited matrix and in the cell lysate. Ideally, the mRNA levels should also be analyzed. These controls are important – otherwise the authors cannot conclude that there are abnormalities in the SECRETION of these proteins.
10. The results shown in Fig. 5 demonstrate reduced granulation tissue formation in zebrafish wounds after treatment with the inhibitors. This is interesting, but again, any membrane-permeable peptide may inhibit granulation tissue formation, e.g. because of toxicity in vivo. The authors should at least check if there is increased apoptosis. The reduced collagen deposition could be a consequence of impaired collagen secretion (expected result), but also of enhanced fibroblast apoptosis (or senescence), of reduced fibroblast proliferation and/or of reduced myofibroblast differentiation. This should be tested. I also recommend analyzing the wound sections using SHG.
11. Fig. 6 lacks Ponceau S staining (or an alternative staining) of the membrane as a loading control. The Western blots look convincing, but it is not clear if any of the differences is statistically significant.

Again, the authors should also analyze the levels of procollagen I, fibrillin and fibronectin in the cell lysate and in the matrix. Otherwise they cannot conclude that SECRETION is inhibited.

12. The authors should comment on how they would apply the peptides to inhibit fibrosis in mammals. A local application to wounds could be feasible, but systemic application is probably problematic.

Minor points:

13. The introduction is rather long and the last part is mainly a repetition of the summary - this could be shortened.

14. Page 19, second paragraph: Results in Materials and Methods should be avoided.

15. Many of the figure legends include a repetition of the results – this should be avoided.

Reviewer #2 (Remarks to the Author):

The manuscript by Raote et al. describes a potential therapeutic strategy for scarring and fibrosis using synthetic peptides to reduce ECM protein secretion, primarily collagens. The authors conducted experiments using 1) HEK293 cells to validate whether the peptides interfere with the interaction between TANGO1 and cTAGE5, 2) zebrafish, two established cell lines, and primary human skin fibroblasts to confirm that the peptides reduce ECM protein secretion, and 3) wound healing and scleroderma models using zebrafish and patient fibroblasts to assess the effects of the peptides on ECM protein secretion.

They demonstrated that the peptides inhibited TANGO1-cTAGE5 heterodimerization, resulting in altered collagen fibril architecture in zebrafish and impaired collagen secretion in the cells. Additionally, they showed that this peptide-based treatment allowed for the control of collagen deposition in zebrafish and the inhibition of collagen secretion in patient-derived primary fibroblasts. The authors concluded that inhibiting ECM protein secretion by interfering with the TANGO1-cTAGE5 binding interface could be a promising approach for therapeutic intervention in fibrotic disorders. This manuscript is expected to be of broad interest to readers. The study's concept is intriguing, the experimental approaches are rational, and the results are presented clearly. However, several concerns need careful consideration.

1) In Fig. 1G, the inhibitory effects by P2, P5 and P2/P5 were quite modest and subtle even at the concentration of 100 uM each, the difference of which does not appear to be statistically meaningful. The authors should consider statistically whether this result is significant.

2) The authors described that inhibiting TANGO1/cTAGE5 heterodimerization by the presence of inhibitory peptides led to unstable monomers which were easily degraded maybe by ER-associated degradation. In this experiment, the syntheses of both proteins, TANGO1 and cTAGE5, were not checked under the presence of inhibitory peptides and the decrease of each protein in the presence of proteasome inhibitors was also not examined. Thus, the validity of the authors' conclusion that TANGO1 and cTAGE5 become unstable without making heterodimers is tenuous on these results.

3) In Fig. 3E and 3F, the addition of P2 and P5 independently did not alter the secretion of collagen I. However, the combination of P2 and P5 at the same concentration caused a marked reduction of collagen secretion (Fig. 3G and 3H). This needs further discussion from the mechanistic aspect. Once one peptide, for example P2, binds its target sequence, it might be sufficient to inhibit the heterodimerization. In addition, the effects on the BRET intensity (Fig. 1G) and on the collagen secretion (Fig. 3G, H) seem to be apparently inconsistent; BRET signal was inhibited by 30% in the presence of 100 uM P2 and P5 (Fig. 1G), whereas collagen secretion was inhibited by 70% in the presence of 40 uM P2 and P5 (Fig. 3H). The effect on the BRET signal might be more sensitive by the treatment, but the results were opposite, which need to be discussed.

4) The authors performed secretome analysis using quantitative mass spectrometry with conditioned

medium from cultured control and P2+P5-treated primary human dermal fibroblasts (Figure 4 and suppl. Figure 6 and 7). In this experiment, fibronectin showed no significant changes between treated and untreated conditions. However, the authors proposed that peptide treatments significantly reduced fibronectin secretion in both healthy and scleroderma fibroblasts (Figure 6). The authors should address this inconsistency between the cells.

5) Immunofluorescence microscopy revealed intracellular collagen accumulation in treated U2OS and RDEB/FB/C7 cells (Suppl. Figure 3). Although peptide treatment clearly impaired collagen I secretion in primary human dermal fibroblasts (Figure 3G and H), no intracellular accumulation of procollagen I was observed in treated fibroblasts (Figure 3C and D). The authors need to address 1) why collagen I does not intracellularly accumulate in primary human dermal fibroblasts, given their statement that "these primary cells can clear most accumulated intracellular collagen" (lines 247-8), and 2) this inconsistency in intracellular collagen accumulation across cell types.

6) Western blotting images for the conditioned media (Figure 3E, 3G, and 6A) need normalization. For example, use a loading control such as GAPDH in Figure 3C or measure the total protein concentration of the conditioned medium. Normalizing the loaded medium by fibronectin, which was unaffected by peptide treatment (Suppl. Figure 7 and comment 1), brings the signal intensity of procollagen I much closer between treated and untreated with the peptides in both healthy and scleroderma fibroblasts (Figure 6A and B).

7) TANGO1's role in collagen secretion has been emphasized by the Malhotra group, one of the corresponding authors of this manuscript. Here, the authors propose that TANGO1 is essential for the secretion of various ECM proteins. Given the authors' expertise in TANGO1 biology and the associated collagen secretory pathway, a more thorough discussion of this expanded role of TANGO1 is expected.

Minor comments:

1) line 147: P2, P5, or P2+P5 (0-200 μ M) / 100 μ M instead of 200 μ M? If 200 μ M is correct, consider including the results of 200 μ M in Figure 1G,

2) Error bars are missing in Figure 1G.

3) line 228-9: We observed increased collagen I (magenta) in peptide-treated cells, which colocalized with the ER-resident protein Calnexin (green) / colors are reversed.

4) The western blotting images with procollagen I (Goat anti-type I collagen / SouthernBiotech) consistently exhibited multiple bands (Figure 3C, 3E, 3G, and 6A). Annotating the bands indicating which alpha chain (COL1A1 or COL1A2) or processing molecule (with or without non-collagenous domain) would be helpful.

Authors' response to reviewers

Reviewer 1

Raote et al generated membrane-permeable peptides that inhibit the interaction between TANGO1 and cTAGE5, resulting in the inhibition of collagen formation. This effect was validated in cultured fibroblasts from healthy individuals and from scleroderma patients. The peptide inhibitors also reduced granulation tissue formation during wound healing in zebrafish. These results suggest the use of the peptide inhibitors for the treatment of excessive scar formation and fibrosis. The role of TANGO1 and cTAGE5 in collagen secretion is not novel, but the development of the peptide is innovative and the translational relevance is strong. However, there are also some weaknesses of this study as detailed below.

1. The results shown in Fig. 1G are not convincing. There is obviously only an effect of the peptides at the highest concentration, and it is unclear if this reduction is statistically significant. Please include a statistical analysis.

The nanoBRET analysis involves expressing the coiled-coil domains of TANGO1 and cTAGE5 that form a dimer in the cytoplasm in our experimental set up. The aim is to test the hypothesis that peptides would inhibit TANGO1-cTAGE5 dimerization, which could be recorded by a fluorescence-based readout that could be adapted for a high-throughput assay in the future. Unfortunately, the signal to noise ratio is low and highly variable. This is because we use transient transfection of the domains and the inability to control expression level, and the actual nanoBRET signal is generally low. After extensive trials we were unable to improve the signal-to-noise. To avoid misleading the readers, we have decided to remove these data and interpretation from the paper.

2. The Western blots shown in Fig. 1H and 1I look convincing, but the quantitative analysis (Fig. 1J) is not convincing. There is obviously a high variability, and the differences are not statistically significant.

We have now added the statistical test on these quantifications. The effects are variable, but show a significant difference in levels of TANGO1 and cTAGE5 between control and treated cells. This has been included in the figure and legend.

3. There is a general problem with the controls in the in vitro and in vivo assays. The authors used PBS or no treatment as control, but the appropriate control would be a membrane-permeable peptide, which does not inhibit TANGO1. It is well known that membrane-permeable peptides can have toxic effects. Such a control should be used at least for the in vitro experiments.

We now include scrambled peptides (the same amino acids but in a scrambled order) for peptides 2 and 5 as controls (see new figure 3K). These scrambled peptides do not influence collagen secretion even at high concentrations. They also do not demonstrate significant toxicity (see new Fig. 3L).

4. Fig. S2: Why is there reduced survival in some of the controls? Why is there such a high variability between experiments when 4 uM of the peptides are used? Overall, it is unclear how the data shown in this figure should be interpreted.

We agree with the reviewer that the data are difficult to interpret. We have removed these survival curves as the subsequent figure 5 provides a more reliable and quantitative TUNEL staining, which shows that the peptides are non-toxic under these conditions.

5. Fig. 2E: It is unclear if any of the differences are statistically significant.

The distribution of 'dispersion' of collagen fibres as visualized by second harmonic generation microscopy has now been tested for statistical significance using the Kolmogorov–Smirnov test. The figure and legend have been modified accordingly.

6. Please quantify the data shown in Fig. S3.

The data (now in supplementary figure 2) have been quantified, showing an increase in intracellular collagens type I (U2OS) and type VII (RDEB/FB/C7) and tested for statistical significance. The figure and legend have been updated accordingly.

7. Fig. S5: It seems that the peptides even inhibit toxicity please comment on this result.

We have now added data from several independent new experiments and also included scrambled control peptides. There is no statistically significant difference between the controls and all the peptides used (new Fig 3L).

8. Fig. 3E and 3F: Why is the combination of both peptides required to inhibit collagen secretion? Is there also less collagen I in the deposited matrix?

- a) Our data clearly presented evidence demonstrating that the individual peptides 2 or 5 failed to down-regulate collagen secretion significantly (Fig 3E, F). We propose that the combination is active due to their specific binding sites in the TANGO1 – cTAGE5 heterodimer, such that the dual blockade interferes with formation of the dimer and leads to its destabilization. This suggestion is included in the Discussion.
- b) Under the conditions used for the cell culture experiments (20h incubation) only very negligible amounts of ECM are deposited around the cells, whereas the vast majority of the molecules is secreted to the culture medium. Western blot and immunofluorescence analysis by showing a low magnification immunofluorescence image using collagen antibodies, show a virtually complete lack of deposited collagen. Some accumulation of procollagen can be detected clearly in peptide-treated fibroblasts, though the magnitude of the effect can vary (see new suppl. Fig. 4). Please also see our response to Reviewer 2's point 5).

9. The results shown in Fig. 4 are interesting, but there is no further validation of the data. The authors should select 3-4 regulated proteins and check their amounts in the secretome, in the deposited matrix and in the cell lysate. Ideally, the mRNA levels should also be analyzed. These controls are important otherwise the authors cannot conclude that there are abnormalities in the SECRETION of these proteins.

We agree with the reviewer, however, we have already confirmed the regulation of collagen I, fibrillin and fibronectin by western blotting (Fig. 6). We here provide data for collagen XII and collagen VI confirming the mass spectrometry data as information for the reviewer (Fig. 1 for R1 below).

Figure 1. Collagen XII and collagen VI secretion from cells treated with P2+P5 at 40 μ M for 20 h showing a decreased secretion of both proteins. Western blots were done with human dermal fibroblast supernatants from two different donors. Ponceau staining was made for the corresponding protein lysates showing that supernatants were derived from equal numbers of cells.

10. The results shown in Fig. 5 demonstrate reduced granulation tissue formation in zebrafish wounds after treatment with the inhibitors. This is interesting, but again, any membrane-permeable peptide may inhibit granulation tissue formation, e.g. because of toxicity *in vivo*. The authors should at least check if there is increased apoptosis. The reduced collagen deposition could be a consequence of impaired collagen secretion (expected result), but also of enhanced fibroblast apoptosis (or senescence), of reduced fibroblast proliferation and/or of reduced myofibroblast differentiation. This should be tested. I also recommend analyzing the wound sections using SHG.

We have now carried out TUNEL staining and do not observe induced apoptosis in the treated samples. We also counted DAPI positive nuclei and found no difference between treated and untreated samples, indicating that the difference observed in the granulation tissue is due to reduced ECM. We have included these results in the modified Fig. 5.

11. Fig. 6 lacks Ponceau S staining (or an alternative staining) of the membrane as a loading control. The Western blots look convincing, but it is not clear if any of the differences is

statistically significant. Again, the authors should also analyze the levels of procollagen I, fibrillin and fibronectin in the cell lysate and in the matrix. Otherwise they cannot conclude that SECRETION is inhibited.

We agree with the reviewers that normalization of the western blotting is critical. We have therefore introduced several measures to guarantee that the same amount of protein was analyzed for all samples. Since the amount of protein in the supernatant was too low to give a signal in the Ponceau staining, we also use normalization to cell number by showing the protein content of cell lysates (new Fig. 3I), which reflects that comparable numbers of fibroblasts gave rise to the proteins in the supernatants, hence equal volumes of supernatants were used for the analysis of secreted proteins (see enclosed Figure 2 below). Ponceau staining is now provided in Fig. 3E and G, and we have described quantification in detail in Methods.

Ponceau S staining from supernatants for Fig 6A

BCA protein estimation of cell lysates

Figure 2. Ponceau staining of Fig 6A western blots (supernatants). BCA protein estimation for cell lysates is shown to reflect that proteins from equal number of cells were loaded to the wells.

12. The authors should comment on how they would apply the peptides to inhibit fibrosis in mammals. A local application to wounds could be feasible, but systemic application is probably problematic.

We thank the reviewer for this suggestion and have added a paragraph in the discussion. Actually, using skin wound healing as a model is a great advantage since topical application of a peptide solution would be a direct approach reducing potential side effects.

13. The introduction is rather long and the last part is mainly a repetition of the summary - this could be shortened.

Introduction has been shortened and repetitions were removed.

14. Page 19, second paragraph: Results in Materials and Methods should be avoided.

We have modified Material and Methods according to the suggestions.

15. Many of the figure legends include a repetition of the results this should be avoided.

We have shortened the figure legends and have omitted duplications as much as possible.

Reviewer 2

The manuscript by Raote et al. describes a potential therapeutic strategy for scarring and fibrosis using synthetic peptides to reduce ECM protein secretion, primarily collagens. The authors conducted experiments using 1) HEK293 cells to validate whether the peptides interfere with the interaction between TANGO1 and cTAGE5, 2) zebrafish, two established cell lines, and primary human skin fibroblasts to confirm that the peptides reduce ECM protein secretion, and 3) wound healing and scleroderma models using zebrafish and patient fibroblasts to assess the effects of the peptides on ECM protein secretion.

They demonstrated that the peptides inhibited TANGO1-cTAGE5 heterodimerization, resulting in altered collagen fibril architecture in zebrafish and impaired collagen secretion in the cells. Additionally, they showed that this peptide-based treatment allowed for the control of collagen deposition in zebrafish and the inhibition of collagen secretion in patient-derived primary fibroblasts.

The authors concluded that inhibiting ECM protein secretion by interfering with the TANGO1-cTAGE5 binding interface could be a promising approach for therapeutic intervention in fibrotic disorders. This manuscript is expected to be of broad interest to readers. The study's concept is intriguing, the experimental approaches are rational, and the results are presented clearly. However, several concerns need careful consideration.

1) In Fig. 1G, the inhibitory effects by P2, P5 and P2/P5 were quite modest and subtle even at the concentration of 100 uM each, the difference of which does not appear to be statistically meaningful. The authors should consider statistically whether this result is significant.

As described to Reviewer 1, point 1, these analyses have been removed from the manuscript. The nanoBRET experiment involves expressing the coiled-coil domains of TANGO1 and cTAGE5 that form a dimer in the cytoplasm in our experimental set up. The aim was to test TANGO1-cTAGE5 dimerization using a fluorescence-based readout that could be adapted for a high-throughput assay in the future. Unfortunately, the signal to noise ratio is low and highly variable. This is because we use transient transfection of the domains and the inability to control expression level, and the actual nanoBRET signal is generally low. After extensive trials we were unable to improve the signal-to-noise. To avoid misleading readers, we have decided to remove these data and interpretation from the paper.

2) The authors described that inhibiting TANGO1/cTAGE5 heterodimerization by the presence of inhibitory peptides led to unstable monomers which were easily degraded maybe by ER-associated degradation. In this experiment, the syntheses of both proteins, TANGO1 and cTAGE5, were not checked under the presence of inhibitory peptides and the decrease of each protein in the presence of proteasome inhibitors was also not examined. Thus, the validity of

the authors' conclusion that TANGO1 and cTAGE5 become unstable without making heterodimers is tenuous on these results.

We agree with the reviewer that further experiments are essential before identifying the mechanisms behind the degradation of TANGO1 and cTAGE5. We have reduced this suggestion and highlighted that it is a speculative idea. Th

3) In Fig. 3E and 3F, the addition of P2 and P5 independently did not alter the secretion of collagen I. However, the combination of P2 and P5 at the same concentration caused a marked reduction of collagen secretion (Fig. 3G and 3H). This needs further discussion from the mechanistic aspect. Once one peptide, for example P2, binds its target sequence, it might be sufficient to inhibit the heterodimerization. In addition, the effects on the BRET intensity (Fig. 1G) and on the collagen secretion (Fig. 3G, H) seem to be apparently inconsistent; BRET signal was inhibited by 30% in the presence of 100 uM P2 and P5 (Fig. 1G), whereas collagen secretion was inhibited by 70% in the presence of 40 uM P2 and P5 (Fig. 3H). The effect on the BRET signal might be more sensitive by the treatment, but the results were opposite, which need to be discussed.

As the reviewer points out, the data from the NanoBRET assay are confusing and do not contribute to a better understanding of the data. We have removed the assay and the associated interpretations.

We could speculate that the combination of P2 and P5 is considerably more effective than the two when used individually, perhaps because of how they occupy the TANGO1-cTAGE5 interface. Since each of them would bind at a different site along the interacting surface, it is reasonable that the two of them together are far more efficacious.

As the reviewer points out, one cannot compare peptide efficacy across the different processes that are affected – leading to differences in the level of effect observed with the NanoBRET assay vs with secretion. There is a correlation between inhibiting hetero-dimerisation, changing rates of protein degradation, and inhibiting collagen secretion – future studies will be useful in delineating the mechanistic details of this process and quantitatively assessing the contribution from each of these steps. At this stage, we have focussed on the effect of the peptide inhibitors on TANGO1 activity and collagen secretion, and highlighted speculative analyses.

4) The authors performed secretome analysis using quantitative mass spectrometry with conditioned medium from cultured control and P2+P5-treated primary human dermal fibroblasts (Figure 4 and suppl. Figure 6 and 7). In this experiment, fibronectin showed no significant changes between treated and untreated conditions. However, the authors proposed that peptide treatments significantly reduced fibronectin secretion in both healthy and scleroderma fibroblasts (Figure 6). The authors should address this inconsistency between the cells.

We thank Reviewer 2 for pointing out this discrepancy and have carried out several new experiments. Western blot analyses demonstrated a clear reduction of FN in the conditioned medium. This is now based on additional independent experiments and is statistically significant (see enclosed Figure 3 below). In the mass spectrometry experiments we noticed some variation with a reduction of fibronectin in 2 experiments and a borderline result in another experiment. Based on the new data we have now excluded FN from the list of unaltered proteins (suppl. Fig. 7).

Figure 3: Human skin fibroblasts from healthy individuals and scleroderma patients were treated as described for Fig. 6 in the manuscript. Levels of fibronectin in supernatants were determined by western blot. Results show significant reduction following treatment with P2+P5 except in TGF β -treated control fibroblasts which display a high degree of variability.

5) Immunofluorescence microscopy revealed intracellular collagen accumulation in treated U2OS and RDEB/FB/C7 cells (Suppl. Figure 3). Although peptide treatment clearly impaired collagen I secretion in primary human dermal fibroblasts (Figure 3G and H), no intracellular accumulation of procollagen I was observed in treated fibroblasts (Figure 3C and D). The authors need to address 1) why collagen I does not intracellularly accumulate in primary human dermal fibroblasts, given their statement that "these primary cells can clear most accumulated intracellular collagen" (lines 247-8), and 2) this inconsistency in intracellular collagen accumulation across cell types.

This reviewer is correct in pointing out that the intracellular accumulation of collagen is mainly observed in established fibroblast cell lines.

However, we have immunofluorescence data demonstrating some minor intracellular accumulation also in primary human skin fibroblasts (new Suppl. Figure 4). Please also see our response to Reviewer 1's point 8). In addition, we also supply EM data for Reviewer 2 to demonstrate some dilatation of the ER in peptide-treated fibroblasts (see enclosed Fig. 4 below). We also want to point out that the western blot shown in Fig. 3C indicates a light intracellular accumulation although this does not reach statistical significance in the summary of all experiments (Fig. 3D). We have now modified the discussion according to these data and also discuss potential mechanisms explaining the difference between cell lines and primary human skin fibroblasts.

Figure 4: Changes in ultrastructure induced in primary human skin fibroblasts by treatment with P2+P5. RER: yellow arrows; stress fibers: arrowheads; Golgi: black arrows.

6) Western blotting images for the conditioned media (Figure 3E, 3G, and 6A) need normalization. For example, use a loading control such as GAPDH in Figure 3C or measure the total protein concentration of the conditioned medium. Normalizing the loaded medium by fibronectin, which was unaffected by peptide treatment (Suppl. Figure 7 and comment 1), brings the signal intensity of procollagen I much closer between treated and untreated with the peptides in both healthy and scleroderma fibroblasts (Figure 6A and B).

As already stated in the response to Reviewer 1 (comment 11) we agree that standardization of the western blotting is critical. We cannot use Ponceau staining of membranes with proteins from serum-free supernatants because there is no staining (please see Fig. 2 for Reviewer 1 above). Also, BCA protein estimation in supernatants is not applicable because it will pick up the added peptides. We have therefore included total protein data from the protein lysates of the cell layers corresponding to the supernatants analyzed by western blot using equal volumes. The protein levels in cell lysates did not show statistically significant differences between treated cultures and controls (see new Fig. 3I). We also include Ponceau staining of membranes with proteins from the cell layers demonstrating identical protein concentrations (see Fig. 3E and G). We have added a detailed description of the normalization of the western blots to Methods.

7) TANGO1's role in collagen secretion has been emphasized by the Malhotra group, one of the corresponding authors of this manuscript. Here, the authors propose that TANGO1 is essential for the secretion of various ECM proteins. Given the authors' expertise in TANGO1 biology and the associated collagen secretory pathway, a more thorough discussion of this expanded role of TANGO1 is expected.

We have included a section in the discussion highlighting how TANGO1 is required for the secretion of many ECM proteins.

Minor comments

1) line 147: P2, P5, or P2+P5 (0-200 μ M) / 100 μ M instead of 200 μ M? If 200 μ M is correct, consider including the results of 200 μ M in Figure 1G,

This assay has now been removed from the manuscript.

2) Error bars are missing in Figure 1G.

We have now removed the results from Figure 1G – the NanoBRET assay for heterodimerization. As described above in response to point 1 raised by reviewer 1 – unfortunately, the signal to noise ratio is low and highly variable; repeated experiments were not able to resolve this problem. We have removed the assay and its conclusions from the manuscript.

3) line 228-9: We observed increased collagen I (magenta) in peptide-treated cells, which colocalized with the ER-resident protein Calnexin (green) / colors are reversed.

We thank Reviewer 2 for pointing out this discrepancy and have corrected this.

4) The western blotting images with procollagen I (Goat anti-type I collagen / SouthernBiotech) consistently exhibited multiple bands (Figure 3C, 3E, 3G, and 6A). Annotating the bands indicating which alpha chain (COL1A1 or COL1A2) or processing molecule (with or without non-collagenous domain) would be helpful.

We now explain the different bands detected by western blot and the collagen antibody used (see Fig 6A). Depending on the experimental conditions, however, sometimes there is a difference in the resolution of all bands and some bands migrate together as one.

REVIEWER COMMENTS

Reviewer #1 (Remarks to the Author):

The authors have addressed most of the concerns of the reviewers (although the changes have unfortunately not been marked in the revision), and the revised manuscript is clearly improved. However, I still have a few comments:

1. It is not clear if the differences in Fig. 3J are statistically significant. A test was obviously performed (see legend), but there is no information about the significance in the figure.
2. Fig. 3K: It seems that 80 μ M of the scrambled peptides also reduced collagen secretion - please comment.
3. Fig. 3J is mentioned in the text after Fig. 3K and 3L- this is confusing.
4. The title of Fig. 5 should be changed - there is no scar yet at this stage of healing. "Scar tissue" should be replaced by "granulation tissue".
5. The authors now discuss that the peptides could be applied topically in the case of hypertrophic scars or in scleroderma. However, it is unclear how this should work in skin tissue that is covered by epidermis. Injection? Application to open wounds in situations where excessive scarring is a risk would be more appropriate.
6. Supplementary Fig. 4B: Is the difference statistically significant?
7. I suggest including the data shown in Fig. 1 for reviewer 1 (COL12A1 and COL6A3 Western blots) as a supplementary figure - they are relevant and further support the authors' hypothesis.

Reviewer #2 (Remarks to the Author):

The authors have appropriately addressed most of my concerns. However, a new concern has arisen with this revised manuscript. They have omitted Figures 1F and G, which directly tested whether the peptides destabilized the TANGO1 and cTAGE5 heterodimer using the NanoBRET assay. The authors explain that the experiment encountered challenges with a low and inconsistent signal-to-noise ratio, possibly due to 1) transient transfection for expressing the protein domains, 2) difficulties in controlling protein expression levels, and 3) the inherently weak nanoBRET signal. While I acknowledge these challenges, I believe this experiment is critical for validating their hypothesis and represents the innovation that underpins the novel concept of this manuscript. As a reviewer, I think it is not advisable to remove these results, which were previously questioned, unless there are solid reasons. This is particularly important given that all other results demonstrating translational relevance are strong and convincing.

NCOMMS-23-34189A-23-34189A

Authors' response to reviewers

Response to Reviewer #1:

The authors have addressed most of the concerns of the reviewers (although the changes have unfortunately not been marked in the revision), and the revised manuscript is clearly improved. However, I still have a few comments:

We have marked all changes in the new version in color.

1. It is not clear if the differences in Fig. 3J are statistically significant. A test was obviously performed (see legend), but there is no information about the significance in the figure.

We modified Fig. 3J by adding asterisks for the significant difference between Ctrl and 24h. The other time points (48 and 72 h) did not differ significantly from Ctrl.

2. Fig. 3K: It seems that 80 μ M of the scrambled peptides also reduced collagen secretion - please comment.

We did densitometric evaluation of band intensities of collagen I bands, which showed a minor change in the scr samples treated with 80 μ M in comparison to Ctrl, but the difference did not reach statistical significance. Possibly the very high concentration of the scrambled peptides could generally affect the cells, although no increased toxicity was noted in our assays. Nevertheless, such high peptide concentration was not used in any of the described inhibitory assays.

3. Fig. 3J is mentioned in the text after Fig. 3K and 3L- this is confusing.

We have modified the text such that the figure panels are mentioned in the correct order.

4. The title of Fig. 5 should be changed - there is no scar yet at this stage of healing. "Scar tissue" should be replaced by "granulation tissue".

We fully agree with R1, and the title has now been changed to "granulation tissue" as suggested.

5. The authors now discuss that the peptides could be applied topically in the case of hypertrophic scars or in scleroderma. However, it is unclear how this should work in skin tissue that is covered by epidermis. Injection? Application to open wounds in situations where excessive scarring is a risk would be more appropriate.

Application to open wounds is certainly also an option and we extended this paragraph in Discussion. The text has been modified:

Skin as a target organ in scleroderma and other sclerosing skin diseases offers to use peptide application as local injections infiltrating the involved tissue. In some situations, e.g. to reduce

excessive scar formation following injury even topical applications allowing direct penetration to the dermal compartment might reduce potential side effects.

6. Supplementary Fig. 4B: Is the difference statistically significant?

In Suppl. Fig. 4, data from 2 independent experiments with 2 different fibroblast strains are demonstrated. The significance between treated and untreated cells is shown in Suppl. Fig. 4B. Taken all data together, intracellular accumulation of collagen is significantly, but mildly increased in treated cells. This was added to the text.

7. I suggest including the data shown in Fig. 1 for reviewer 1 (COL12A1 and COL6A3 Western blots) as a supplementary figure - they are relevant and further support the authors' hypothesis.

As suggested, we included the former Figure 1 for R1 as new Supplementary figure 6.

Response to Reviewer #2:

The authors have appropriately addressed most of my concerns. However, a new concern has arisen with this revised manuscript. They have omitted Figures 1F and G, which directly tested whether the peptides destabilized the TANGO1 and cTAGE5 heterodimer using the NanoBRET assay. The authors explain that the experiment encountered challenges with a low and inconsistent signal-to-noise ratio, possibly due to 1) transient transfection for expressing the protein domains, 2) difficulties in controlling protein expression levels, and 3) the inherently weak nanoBRET signal. While I acknowledge these challenges, I believe this experiment is critical for validating their hypothesis and represents the innovation that underpins the novel concept of this manuscript. As a reviewer, I think it is not advisable to remove these results, which were previously questioned, unless there are solid reasons. This is particularly important given that all other results demonstrating translational relevance are strong and convincing.

We hesitate to include data from a system that has not yielded reliable results. The NanoBRET assay encountered challenges with a low and inconsistent signal-to-noise ratio, possibly due to transient transfections variable expression levels and low NanoBRET signal. In addition, several other interactions will influence the interaction of TANGO1 and cTAGE5, which are not replicated in the short coiled-coil-based analysis performed. The effects of peptides on TANGO1/cTAGE5 degradation and on wound healing are consistent and efficacious and stand alone, given their timely nature and the magnitude of the challenges posed by fibrotic disorders. We think it is important to focus on these features and delve into alternate in-depth experiments to get at more detailed mechanistic analyses for a follow-up study.

REVIEWERS' COMMENTS

Reviewer #1 (Remarks to the Author):

The authors have well addressed my remaining concerns.